# A rare *CTSC* mutation in Papillon-Lefèvre Syndrome results in abolished serine protease activity and reduced NET formation but otherwise normal neutrophil function

Felix P. Sanchez Klose[1]*, Halla Björnsdottir[1], Agnes Dahlstrand Rudin[1], Tishana Persson[1], Arsham Khamzeh[1], Martina Sundqvist[2], Sara Thorbert-Mros[3], Régis Dieckmann[2], Karin Christenson[1], Johan Bylund[1]

1 Department of Oral Microbiology and Immunology, Institute of Odontology, Sahlgrenska Academy at University of Gothenburg, Gothenburg, Sweden, 2 Department of Rheumatology and Inflammation Research, Institute of Medicine, Sahlgrenska Academy at University of Gothenburg, Gothenburg, Sweden, 3 Specialist Clinic of Periodontics, Gothenburg, Public Dental Service, Region Västra Götaland, Sweden

* Felix.Klose@gu.se

**Data Availability Statement:** The results of the genetic analysis have been submitted to the

## Abstract

Papillon-Lefèvre Syndrome (PLS) is an autosomal recessive monogenic disease caused by loss-of-function mutations in the *CTSC* gene, thus preventing the synthesis of the protease Cathepsin C (CTSC) in a proteolytically active form. CTSC is responsible for the activation of the pro-forms of the neutrophil serine proteases (NSPs; Elastase, Proteinase 3 and Cathepsin G), suggesting its involvement in a variety of neutrophil functions. In PLS neutrophils, the lack of CTSC protease activity leads to inactivity of the NSPs. Clinically, PLS is characterized by an early, typically pre-pubertal, onset of severe periodontal pathology and palmoplantar hyperkeratosis. However, PLS is not considered an immune deficiency as patients do not typically suffer from recurrent and severe (bacterial and fungal) infections. In this study we investigated an unusual *CTSC* mutation in two siblings with PLS, a 503A>G substitution in exon 4 of the *CTSC* gene, expected to result in an amino acid replacement from tyrosine to cysteine at position 168 of the CTSC protein. Both patients bearing this mutation presented with pronounced periodontal pathology. The characteristics and functions of neutrophils from patients homozygous for the 503A>G *CTSC* mutation were compared to another previously described PLS mutation (755A>T), and a small cohort of healthy volunteers. Neutrophil lysates from patients with the 503A>G substitution lacked CTSC protein and did not display any CTSC or NSP activity, yet neutrophil counts, morphology, priming, chemotaxis, radical production, and regulation of apoptosis were without any overt signs of alteration. However, NET formation upon PMA-stimulation was found to be severely depressed, but not abolished, in PLS neutrophils.

European Genome-phenome Archive and access can be granted with the EGA ID: EGAS00001005040 or with the help of the link: https://ega-archive.org/studies/EGAS00001005040.

**Funding:** The study was supported by the Swedish Research Council ([http://www.vr.se]; 2016-00982 & 2019-01123), the Swedish Heart-Lung Foundation ([http://www.hjart-lungfonden.se]; 20180218), the Patent Revenue Fund Research in Preventive Odontology (www.pmf.se), and by grants from TUA Research Funding; The Sahlgrenska Academy at University of Gothenburg / Region Västra Götaland, Sweden (TUAGBG-917531). All funding was awarded to JB. The funders had no role in study design, data collection and analysis, decision to publish, or preparation of the manuscript.

**Competing interests:** The authors have declared that no competing interests exist.

## Introduction

Neutrophil granulocytes are phagocytic white blood cells that are critical components of the inflammatory process and fight against invading pathogens. They are loaded with a wide variety of antimicrobial compounds and proteolytic enzymes that contribute to microbial killing but also risk damaging endogenous tissues if not properly regulated [1]. Among these enzymes, the neutrophil serine proteases (NSPs) are three structurally related enzymes, Human Neutrophil Elastase (HNE), Proteinase 3 (PR3), and Cathepsin G (CTSG) [2]. In the human population, a number of hereditary symptoms exist that affect neutrophil function and most of these are more or less severe immune deficiencies characterized by recurrent and severe bacterial and fungal infections. In addition, some immune deficiencies are characterized by an inability to regulate inflammation. For instance, patients with chronic granulomatous disease (CGD) are not only plagued by infections but also by a variety of, often aseptic, inflammatory disorders [3, 4]. This indicates that neutrophils are not only critical microbial killers but that they are also important regulators of inflammatory processes.

One hereditary neutrophil defect is the rare Papillon-Lefèvre Syndrome (PLS) with an estimated prevalence of one to four persons per million in the general population [5]. PLS is caused by loss-of-function mutations in the *CTSC* gene resulting in the lack of enzymatic activity of the exo-cysteine protease Cathepsin C (CTSC) [6]. This protease is critical for the activation of the proforms of NSPs in neutrophils, thus PLS neutrophils are deficient in NSP activity [7]. The NSPs are all formed as enzymatically inactive zymogens; after proteolytic removal of two N-terminal amino acids by CTSC they are stored as active granule enzymes during the promyelocyte stage [2, 8]. The activity of NSPs has been thought to be vital for the ability of neutrophils to perform microbial killing and to degrade microbes. PLS neutrophils have been shown to kill bacteria *in vitro* in a similar capacity to neutrophils from healthy controls [9]. Additionally, PLS neutrophils have been found to have less directional chemotactic accuracy to fMLF as well as reduced capacity to produce neutrophil extracellular traps (NETs) upon PMA stimulation [10, 11]. Despite NSP activity being widely held as important enzymes for the killing and degradation of phagocytosed microbes, individuals with PLS are typically not plagued by recurring opportunistic infections. The major clinical finding in PLS is a remarkably aggressive form of periodontitis with manifestation already in the primary dentition, which may lead to complete edentulousness in young adults [12]. Periodontitis is not a typical infectious disease, but rather a chronic inflammatory condition of the tooth-supporting tissues. The condition is triggered by oral bacteria residing in the gingival pockets and it is characterized by progressive destruction of tooth supporting structures [13, 14].

Multiple different mutations in the *CTSC* gene may result in PLS and it is unclear whether different mutations result in different levels of disease severity or altered neutrophil function. Previous reports on PLS neutrophils have demonstrated normal morphology and differential counts of circulating white blood cells [9], reduced production of NETs triggered with phorbol myristate acetate (PMA) [9,10], defective chemotaxis and increased PMA induced reactive oxygen species (ROS) production [10]. Residual CTSC and NSP activity has also been reported for certain patients with PLS [15, 16].

In this report, we describe an unusual 503A>G mutation in *CTSC* in two siblings with PLS (family A) and characterize several basic neutrophil functions with special emphasis on CTSC and NSP activity. These findings were compared with data from two other patients with PLS (siblings of family B) with a previously described *CTSC* mutation (755A>T), as well as from a small cohort of healthy individuals. We found that the 503A>G *CTSC* mutation resulted in abolished CTSC and NSP activity, quite similar to the 755A>T mutation. The basic neutrophil functions investigated were largely found to be within the range displayed by healthy controls,

apart from PMA-triggered NET formation that was potently reduced and delayed, but not completely absent, in PLS neutrophils.

## Patients and methods

### Clinical description

Four patients, treated at the Specialist Clinic of Periodontics, Region Västra Götaland, Gothenburg, Sweden, from two families (A and B) with PLS were recruited for this study.

The two siblings from family A were at the time of the most recent sampling 32 and 28 years old. They had experienced periodontitis since early childhood with a rapid loss of tooth supporting structures and teeth. Both siblings became edentulous in their teens and the older sibling was restored with implants. All implants were then affected by peri-implantitis, and at the time of sampling >50% of the bone support was lost, and the implants in the upper jaw were scheduled for removal and have since been removed. The younger sibling remained edentulous. Another two siblings in family A were clinically healthy. The siblings were of Iraqi descent with parents who were closely related, consanguine.

Family B was of ethnic Somali descent, but the children were born in Sweden. No known consanguinity was reported between the parents. The patients of family B were 11 and 9 years old at the time of sampling and had been in specialist care for periodontal pathology since the age of two. Both had lost all their deciduous teeth and exhibited severe bone loss at the recently erupted permanent teeth.

In addition to periodontal pathology, all patients displayed the PLS-typical palmoplantar localized hyperkeratosis. Besides these classical symptoms, all four were systemically healthy, except that one patient in family B had experienced an episode of a purulent skin cyst, which had to be surgically removed.

### Blood samples

The study was approved by the Regional Ethical Review Board in Gothenburg, Sweden (Dnr: 118–16, 544–17). Peripheral blood was sampled through venipuncture and collected in sample tubes with anti-coagulant (Heparin, or EDTA for genetic analysis). Samples from patients as well as healthy controls without periodontitis were collected after written informed consent was given by the individuals or their legal guardians. For the healthy controls, twelve volunteers aged between 25 and 55 years old were recruited among personnel and students at the at the Institute of Odontology, Sahlgrenska Academy at University of Gothenburg. All attended regular dental checkups, and none reported periodontitis.

### Genetic sequencing and analysis

For family A, the DNA was extracted from whole blood samples collected in EDTA tubes with the help of a GenElute Blood Genomic DNA kit (# NA2010-1KT, Sigma-Aldrich, St Louis, US-MO). Whole genome sequencing was performed at the Core Facilities, Sahlgrenska Academy, University of Gothenburg, Sweden. Afterwards, the data was processed with the help of the Bioinformatics Core Facility, Sahlgrenska Academy, University of Gothenburg, Sweden. First the quality of the data was assessed with FastQC (version 0.11.2) and then Samtools (version 1.3.1) was used to sort, index and assess the mapping statistics. Paired end reads were aligned to the human reference genome hg19 (GRCh37, RefSeq assembly accession GCF_000001405.13) using a Burrows-Wheeler Aligner (BWA mem version, BWA 0.7.13) [17]. Duplicates were removed with Picard (version 2.2.4). With the help of the Genome Analysis ToolKit (GATK, version 3.1–1) realignment and variant calling was performed [18]. With

the Haplotype Caller tool, the variant calling was performed according to GATK best practice, applying the following quality filters for SNPs: QD < 2.0, MQ < 40.0, FS > 60.0, ReadPos-RankSum < -8.0, MQRankSum < -12.5, and for indels: QD < 2.0, FS > 200.0, ReadPosRank-Sum < -20.0. With the known gene database the variants were annotated by using the ANNOVAR tool [19]. Additionally, ANNOVAR was used to annotate exonic variants with functional predictions with the help of the following tools: whole-exome SIFT [20], PolyPhen2 HDIV [21], PolyPhen2 HVAR [21], LRT [22], MutationTaster [23], MutationAssessor [24], FATHMM [25], MetaSVM [26], MetaLR [26], VEST [27], CADD [28], GERP++ [29], ClinVar [30], PhyloP [31, 32] and SiPhy scores [33] from dbNSFP (version 2.6) [34].

For family B, blood samples collected in EDTA tubes were whole exome sequenced and analyzed (as part of a clinical investigation) at the Department of Clinical Genetics at Sahlgrenska Hospital, Gothenburg, Sweden.

## Neutrophil isolation

Neutrophils were isolated from peripheral blood samples of patients and healthy controls based on the isolation method described by Bøyum *et al.* [35]. In short, with the help of dextran, the erythrocytes were precipitated and a Ficoll-Paque gradient was used to separate and then remove the peripheral blood mononuclear cells (PBMCs) from the granulocytes. Remaining erythrocytes in the granulocyte fraction were lysed with distilled water. After washing with Krebs Ringer phosphate buffer (KRG) the samples were diluted in KRG with Calcium (1 mM) and kept at 4°C for further use.

## Neutrophil lysates

Neutrophils isolated from peripheral blood samples were diluted in KRG with Calcium (1 mM) to $5x10^6$ cells/ml, pelleted and lysed in 0.1% Triton X-100. Vortexing and centrifugation at 15871 *rcf* for 25 min at 4°C was used to mechanically lyse the cells. Afterwards, the lysates were diluted in KRG with Calcium (1 mM) and the cellular debris was removed by centrifugation (short mode, max. 21,130 *rcf*, 30 s, 4°C). The resulting samples, all at a concentration of $2.5x10^6$ cell equivalents/ml, were stored at -80°C.

## Neutrophil cytospins

The isolated neutrophil samples were diluted to a concentration of $1x10^6$ cells/ml and then fixed on a microscope slide through cyto-centrifugation at 130 *rcf* for 5 min (THARMACspin CS1, Waldsolms, Germany). After drying, the microslides were stained with Giemsa and May-Grünwald staining (both from Sigma-Aldrich) and micrographs were taken at 100x magnification with a microscope (Objective: Olympus UPlan FL U 100x/1.30 Oil Ph3 ∞/0.17/FN26.5, Microscope: BX41, LRI Olympus, Camera: Olympus DP71, Tokyo, Japan).

## Neutrophil activation assay and flow cytometry

Whole blood samples (540 µl) were supplemented with TNFα (#T6674, Sigma-Aldrich) to a final concentration of 10 ng/ml or left unstimulated. After incubation at 37°C for 20 min, the samples were transferred into FACS Lysing solution (diluted 1:10 in dH2O, #349202, BD Biosciences, Franklin Lakes, US-NJ), vortexed and placed on ice for 15 min before washing with PBS (277 *rcf*, 10 min, 4°C). After aspiration of the supernatant, the cells were resuspended in FACS Lysing solution (1:10, BD Biosciences) and the incubation and washing steps were repeated as previously. The cell pellets were resuspended in PBS and antibodies (all from BD Biosciences) were added in accordance with the dilutions recommended by the manufacturer;

1:20 for CD35 PE (#559872) and CD11b APC (#333143), and 1:40 for CD62L PE (#341012). One control per donor sample was left unlabeled. The samples were incubated in the dark at 4°C for 30 min and washed one more time in PBS before measurement in a flow cytometer (Accuri C6, BD Biosciences). The results were analyzed with the help of FlowJo 10 (BD Biosciences).

## Protease activity

Protease activity was assessed in a plate reader (BMG Labtech, Ortenberg, Germany) with fluorogenic substrates; H-GR-AMC hydrochloride salt (# I-1215.0050, Bachem, Bubendorf, Switzerland) for CTSC activity, Abz-APEEIMRRQ-EDDnp (# SNE-3230-v, Peptides International, Louisville, US-KY) for HNE activity, Abz-VADXRDR-EDDnp (# SNP-3232-v, Peptides International) for PR3 activity, and Abz-EPFWEDQ-EDDnp (# SFR-3231-v, Peptides International) for CTSG activity. The NSP substrates are currently available from vivitide (www. vivitide.com).

The samples were diluted in KRG with Calcium (1 mM), added to black 96 well culture plates (Thermo Scientific, Waltham, US-MA) and incubated at 37°C for 15 min. Afterwards, the pre-incubated samples were mixed 1:1 with the respective substrate-buffer mix to their final substrate-specific concentration as shown in Table 1. Cell equivalent normalization was used to achieve comparability between the lysates. The fluorescence intensity (FI) of each sample was measured in triplicate every 3 minutes for at least 30 minutes under regular shaking of the plate and at the fluorophore-specific wavelengths according to Hamon *et al*. and Attucci *et al*. [36, 37] (Table 1). The design of the protease activity system was based on a system developed by Korkmaz *et al*. [38]. Briefly, the substrates were provided in abundance to avoid saturation of the fluorescence increase and allow for a longer linear increase in FI. Linear regression was performed on the change of FI over time for the linear segment of the kinetic

**Table 1. Parameters for protease activity assay.**

| Protease | Substrate concentration (µM) | Buffer | | Lysate dilution | Measurement (nm)[a] | |
|---|---|---|---|---|---|---|
| | | Material | Concentration | | | |
| Cathepsin C (CTSC) | 1500.0 | Na Acetate | 50 mM | 1:50 | Exc.: | 355±15 |
| | | NaCl | 30 mM | | Em.: | 460±20 |
| | | EDTA | 10 mM | | | |
| | | DTT | 2 mM | | | |
| Neutrophil Elastase (HNE) | 5.0 | HEPES | 50 mM | 1:20 | Exc.: | 320±15 |
| | | NaCl | 750 mM | | Em.: | 420±20 |
| | | Igepal CA-630 | 0.05 vol% | | | |
| | | pH | 7.4 | | | |
| Proteinase 3 (PR3) | 5.0 | HEPES | 50 mM | 1:300 | Exc.: | 320±15 |
| | | NaCl | 750 mM | | Em.: | 420±20 |
| | | Igepal CA-630 | 0.05 vol% | | | |
| | | pH | 7.4 | | | |
| Cathepsin G (CTSG) | 20.0 | HEPES | 50 mM | 1:10 | Exc.: | 320±15 |
| | | NaCl | 100 mM | | Em.: | 420±20 |
| | | Igepal CA-630 | 0.01 vol% | | | |
| | | pH | 7.4 | | | |

[a]Filter settings for excitation (Exc.) and emission (Em.) wavelength for the fluorescence measurement of the substrates. Peak wavelength±range.

curve (0–12 min). The calculated slope of the linear regression was then normalized according to the cell equivalents used, which in turn can be related to the amount of protease activity in the sample, allowing for comparison of the protease activity in samples from different individuals.

## Immunoblotting for CTSC

The presence of CTSC was tested through immunoblotting of samples prepared from isolated neutrophils. The neutrophils (50 μl, $5x10^6$ cells/ml) were mixed with $dH_2O$, LDS Sample Buffer (Invitrogen, Carlsbad, US-CA) and Reducing Agent (Invitrogen). Triton X-100 was added to a final concentration of 0.5% and AEBSF (Thermo Scientific) to a final concentration of 1 mM. The samples were vortexed for 30 s, boiled at 90˚C for 5 min and stored at -80˚C. Recombinant human CTSC (rhDPPI; UNIZYME Laboratories A/S, Horsholm, Denmark) was mixed with $dH_2O$, LDS Sample Buffer as a control and boiled together with the frozen samples at 90˚C for 5 min. Samples, normalized on basis of cell number (each lane was loaded with 31,250 cell equivalents and the CTSC control with 1.0 ng/lane), were then separated on a Bolt bis-tris plus 4–12% gel (Invitrogen) in a mini gel tank electrophoresis bath (Invitrogen) filled with MES SDS running buffer (Invitrogen) with a PowerEase300W power source (Invitrogen) and blotted on a transfer membrane (Invitrogen) in an iBlot2 dry blotting system (Life Technologies, Waltham, US-MA). The membrane was blocked with 5% (w/v) BSA in 0.05% TWEEN in PBS and subsequently incubated with the mouse monoclonal Anti-cathepsin C Antibody (D-6) (1:250; sc-74590, Santa Cruz Biotechnology, Dallas, US-TX) and the secondary Polyclonal Rabbit Anti-Mouse Ig conjugated to HRP (1:500; P 0260, Dako Denmark A/S, Glostrup, Denmark). After labelling with the antibodies, the membranes were developed for 3 min with a WB enhanced chemiluminescence (ECL) kit (Thermo Scientific) and scanned in a Gel scanner (ChemiDoc XRS, Bio-Rad Laboratories, Hercules, US-CA).

## NADPH-oxidase derived reactive oxygen species production

The production of extracellular and intracellular reactive oxygen species (ROS) was measured in a chemiluminescence (CL)-based system [39] as CL activity in a ClarioStar plate reader (BMG Labtech) for family A or Biolumat LB 9505 (Berthold Co., Wildbad, Germany) for family B. The assay was performed in a white 96-well microtiter plate (volume 200 μl) or polypropylene tubes, respectively. For extracellular ROS measurements, neutrophils (100 μl, $5x10^6$ cells/ml) were mixed with Isoluminol (5.64 μM, Sigma-Aldrich) and horseradish peroxidase (HRP, 0.2 U/ml, Roche Diagnostics, Basel, Switzerland). For intracellular measurements, neutrophils (100 μl, $5x10^6$ cells/ml) were mixed with Luminol (5.64 μM, Sigma-Aldrich), superoxide dismutase (SOD, f0.5 U/ml, Worthington Biochemical Corp., Lakewood, US-NJ) and Catalase (20 U/ml, Worthington). After mixing, the samples were pre-incubated for 5 min at 37˚C. A background base line was then established before adding PMA (50 nM, Sigma-Aldrich) to the wells. The ROS production was measured for a total of 20 min or until the bulk of the reaction had concluded.

## PMA-induced NETosis

The NET formation was visualized according to a protocol by Björnsdóttir *et al.* [40]. In short, neutrophils ($2.5x10^5$ cells/coverslip) were allowed to settle on polylysine coated coverslips (Knittel Glass, Braunschweig, Germany) and incubated in cell culture medium supplemented with PMA (50 nM, Sigma-Aldrich) for 3 h at 37˚C and 5% $CO_2$. Control cells were incubated without PMA. The cells were fixed with 4% PFA at room temperature for 30 min and subsequently washed with PBS. The slides were blocked in PBS supplemented with 10% Donkey

Serum (Abcam, Cambridge, UK) and 2% bovine serum albumin (BSA, Sigma-Aldrich) and incubated with rabbit-anti-human MPO (1:2000 in blocking buffer, # A0398, Agilent, Santa Clara, US-CA) for 30 min at room temperature [11]. After washing, the samples were incubated with donkey-anti-rabbit AF647 (1:500 in blocking buffer, # A-31573, Molecular Probes, Waltham, US-MA) for 30 min at room temperature and washed a final time. The coverslips were mounted on slides with mounting medium containing DAPI (# P36935, Invitrogen) and imaged in a microscope (Objective: Olympus PlanApo 40x/0,95 $\infty$/0,11–0,23, Microscope: LRI Olympus BX41, Camera: Olympus DP71).

To quantify NETosis, neutrophils were suspended in phenol red-free RPMI culture medium (Gibco, Waltham, US-MA) supplemented with Sytox Green DNA stain (1.25 μM, Molecular Probes). The mixture was placed in black 96 well culture plates (Thermo Scientific) at a concentration of $5x10^4$ cells/well. PMA (Sigma-Aldrich) was added to some wells at a concentration of 50 nM. The plates were incubated for 4 h at 37°C and 5% $CO_2$. The fluorescence (excitation 485 nm, emission 535 nm) was measured at indicated time-points in a ClarioStar plate reader (BMG Labtech) [41].

## Neutrophil apoptosis

The regulation of apoptosis was investigated as described by Christenson *et al.* [42]. In short, the neutrophils were suspended in RPMI cell culture medium (Gibco) supplemented with 10% Fetal Bovine Serum (GE Life Sciences, Chicago, US-IL) and 1% penicillin-streptomycin (PEST, Life Technologies) at a concentration of $5x10^6$ cells/ml. The cell suspension was split into different tubes and supplemented with either purified anti-human CD95 (Fas) antibody (10 μg/ml, # 305704, BioLegend, San Diego, US-CA), lipopolysaccharides (LPS) from *E. coli* (100 ng/ml, # L2880, Sigma-Aldrich) or left unstimulated. The neutrophils were incubated at 37°C, 5% $CO_2$ for 20 h.

After incubation, samples were centrifuged at 277 *rcf* for 10 min and the pellet washed with 2 ml Annexin V-Buffer (1 mM HEPES, 14 mM NaCl, 250 μM $CaCl_2$) and centrifuged again. The supernatant was replaced with a cell staining solution of 105 μl Annexin V-Buffer, 1.5 μl Annexin-V-Fluos (Roche Diagnostics) and 5 μl 7-Aminoactinomycin D (7-AAD, BD Biosciences). The mix was incubated at room temperature in the dark for 10 min before cells were immediately analyzed in a flow cytometer (Accuri C6, BD Biosciences). Neutrophils that stained positive for Annexin-V but negative for 7-AAD were classified as apoptotic and the percentage of apoptotic cells in the sample was calculated with the number of total cell events.

## Chemotaxis

Isolated neutrophils were resuspended in KRG with Calcium (1 mM) and 0.3% BSA to a final concentration of $2x10^6$ cells/ml. 30 μl of the cell suspension ($6x10^4$ cells) were loaded on top of a 3-μm polycarbonate chemotaxis membrane with a hydrophobic mask (ChemoTX Disposable Chemotaxis System, Neuro Probe Inc., Gaithersburg, US-MD). The neutrophils were allowed to migrate through the membrane towards buffer, KRG with Calcium (1 mM) and 0.3% BSA, and buffer supplemented with 10 nM fMLF into the provided clear 96 well plate inside an incubator at 37°C, 5% $CO_2$ for 90 min. After lysing the cells with 2% cetyltrimethylammonium in PBS with 2% BSA, the myeloperoxidase activity was measured as absorbance at 450 nm with the help of a peroxidase reagent (OPD, Sigma-Aldrich) in phosphate citrate buffer (0.05 M, pH 5) in a plate reader (CLARIOstar, BMG Labtech). All samples were run as triplicates and normalized to control wells with the maximum number of cells ($6x10^4$ cells).

## Statistical analysis

Statistical analysis of the data was performed with Prism 8.0 (GraphPad Software, San Diego, US-CA). Values of $p < 0.05$ were regarded as statistically significant. The specific statistical tests used are stated were applicable.

## Results

### Genetic and initial findings

A genetic analysis was performed, and the patients of family A (PLS 1 and 2) were found homozygous for the gene variant GCF_000001405.13(CTSC):g.88042469T>C, c.503A>G, p. (Tyr168Cys) (submitted to the European Genome-phenome Archive and stored with the EGA ID: EGAS00001005040). This is a variation (rs750898600 in the ensemble database) at position 88042469 (according to the reference genome hg19) leading to a 503A>G substitution in exon 4 of the *CTSC* gene, which is expected to result in an amino acid replacement from tyrosine to cysteine at position 168 of the CTSC protein. This substitution was predicted to be deleterious for protein function by the Polyphen2 HDIV, Polyphen2 HVAR, LRT, MutationTaster and FATHMM method, among others. This mutation is rare with only sporadic heterozygote individuals reported, giving rise to different allele frequencies in different databases; gnomAD: C = 0.000032/1 [43], TOPMed: C = 0.000016/2 [44], and no reports in ALFA [45]. We only found this genetic variant entered in genetic databases for heterozygous carriers, and to the best of our knowledge, data of only one patient with PLS homozygous for 503A>G has been previously published by Sørensen *et al.* [9] without submission to genetic databases. It was not possible for us to genotype unaffected relatives in family A.

PLS 3 and 4 of family B were whole exome analyzed for diagnostic purposes and were both homozygous for the variant NM_001814.5:c.755A>T, p.(Gln252Leu), resulting in a 755A>T transversion in exon 5 of the *CTSC* gene (rs104894207 in the ensemble database). This mutation is identical to that of patients of a family described in both Toomes *et al.* and Hewitt *et al.* [7, 46] (Table 2) and it is predicted to result in an amino acid change from glutamine to leucine in position 252 of the CTSC protein. In addition, the mother of PLS 3 and 4 participated in the genetic analysis and was heterozygous for the same variant.

Peripheral blood samples from all patients displayed apparently normal leukocyte counts with normal levels of circulating neutrophils that presented with normal size, morphology and granularity as based on microscopy and flow cytometry (Fig 1A and 1B).

### Cathepsin C; activity and presence

Neutrophils isolated from peripheral blood samples of the patients in family A were used to prepare lysates in order to assess enzymatic activity of CTSC. We included neutrophil lysates

**Table 2. Overview of the PLS patients.**

| Patients | | Dental status/diagnosis | Mutation | |
|---|---|---|---|---|
| | | | HGVS[a] | SNP[b] |
| Family A | PLS 1 | Edentulous | GCF_000001405.13(CTSC): | rs750898600 |
| | PLS 2 | Peri-implantitis | g.88042469T>C, c.503A>G, p.(Tyr168Cys) | |
| Family B | PLS 3 | Periodontitis | NM_001814.5(CTSC): | rs104894207 |
| | PLS 4 | | c.755A>T, p.(Gln252Leu) | |

[a]Nomenclature of the Human Genome Variation Society (HGVS);
[b]Single nucleotide polymorphism (SNP).

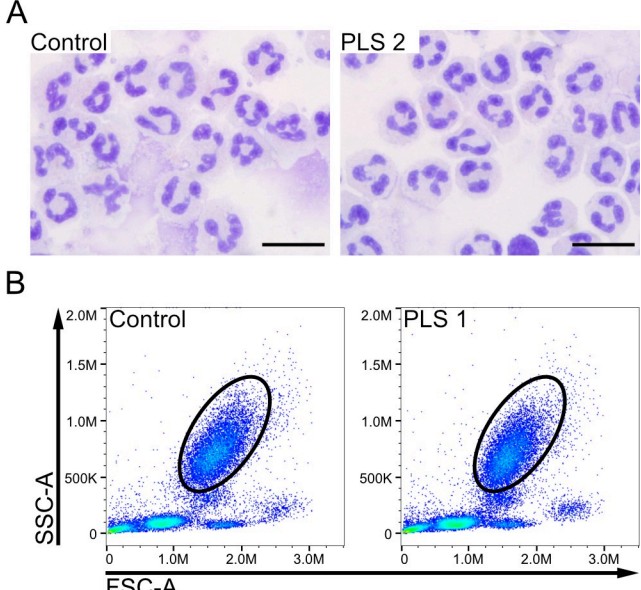

**Fig 1. Cytospins of isolated neutrophils and flow cytometry scatter plots of leukocytes samples.** (A) Representative micrographs of isolated neutrophils from PLS 2 and a healthy control. Neutrophils were fixed on glass coverslips with a cytocentrifuge and stained with Giemsa and May-Grünwald staining. Magnification, 100x. Scale bars, 10 μm. (B) Whole blood samples were treated with FACS lysing solution and measured in a flow cytometer. Presented here through forward- and side-scatter for size and granularity, respectively. The gates highlight the granulocyte population in the samples.

from family B (n = 2) as well as those from a small cohort of healthy control donors (n = 12) with no history of periodontal problems.

A fluorescence-based activity assay was used to measure the activity of CTSC where an enzyme-specific substrate (H-GR-AMC) was exposed to neutrophil lysates; the proteolytic activity of CTSC cleaves this substrate to generate free AMC, the fluorescence of which was measured over time (Fig 2A). Neutrophil lysates from family A were completely devoid of CTSC activity (Fig 2B), indicating that the 503A>G mutation is a complete loss-of-function mutation. Similarly, CTSC activity was completely absent in neutrophil lysates from family B. Among healthy donors, a pronounced interindividual variation in CTSC activity was found even when all lysates were assayed side-by-side on a single plate (Fig 2B).

Having showed the absence of CTSC activity, we wanted to investigate whether CTSC protein could be found in the neutrophils of family A. Immunoblotting for CTSC demonstrated that neutrophils from both patients of family A lacked CTSC, whereas control samples displayed clear bands at the expected molecular weight, around 23kD (Fig 3).

## Activities of downstream serine proteases

We next quantified the enzymatic activity of the NSPs downstream of CTSC using the following substrates: Abz-APEEIMRRQ-EDDnp for HNE, Abz-VADXRDR-EDDnp for PR3, and Abz-EPFWEDQ-EDDnp for CTSG. Like our results for CTSC activity (Fig 2B), levels of neutrophil NSP activity were highly variable within our cohort of healthy donors (Fig 4). Cell lysates from all patients with PLS lacked NSP activity and there were no apparent differences between samples from family A and B (Fig 4).

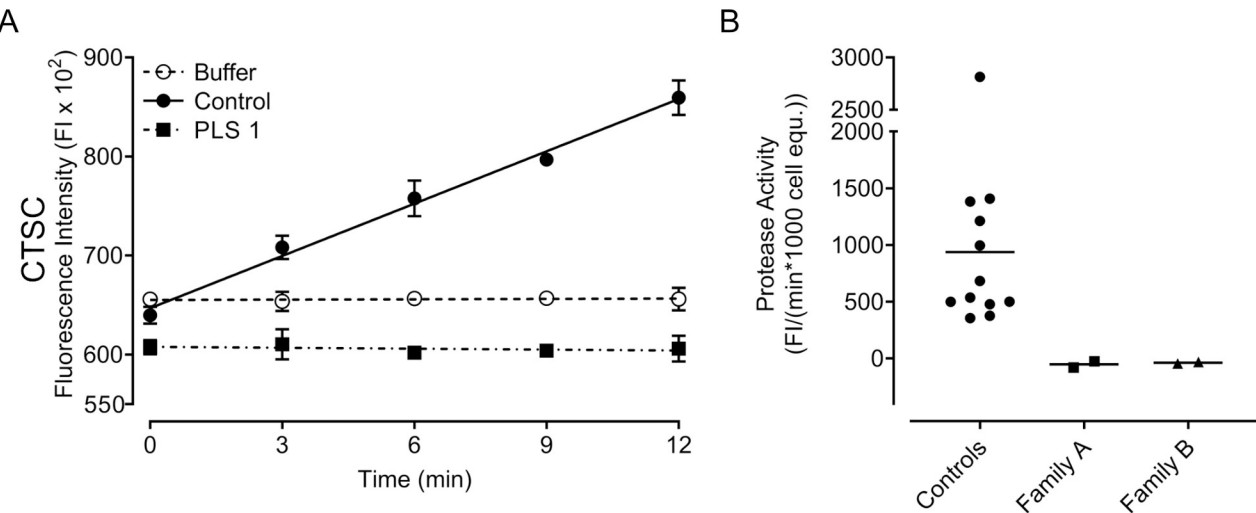

**Fig 2. Fluorescence-based protease activity assay for CTSC activity of neutrophil lysates.** (A) A representative result of CTSC activity assay with lysates from neutrophil samples from a healthy control, PLS 1, and a lysate-free buffer control. Symbols are means ± standard deviations (of three technical repeats) and lines represent the result of the linear regression. (B) Protease activities in neutrophils of patients with PLS (family A, n = 2 and family B, n = 2) and healthy controls (n = 12) resulting from the linear regression of the fluorescence measurement on the linear section of the kinetic graph (0–12 min). Each symbol represents mean per individual (of three technical repeats) and the horizontal line represents the mean of all samples from the indicated groups.

### Functional assessment of PLS neutrophils

Having confirmed that PLS neutrophils with the 503A>G *CTSC* mutation indeed lacked NSP activity, we next wanted to explore whether this would entail any alterations in basic neutrophil functions. Therefore, neutrophils of patients from family A were tested in a variety of functional *in vitro* assays. The neutrophils were primed with the help of TNFα and

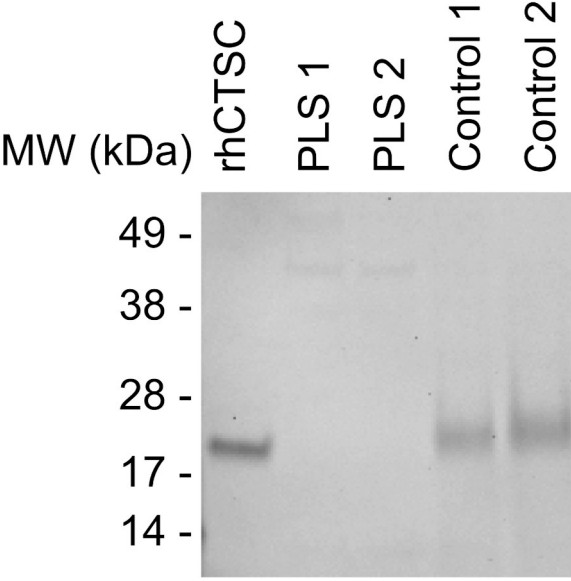

**Fig 3. Immunoblot of CTSC on neutrophil lysates of family A and two healthy controls.** Neutrophil lysates (31,250 cell equivalents/sample) from patients of family A and two healthy controls are shown. Recombinant CTSC (1.0 ng/lane) is shown as a positive control. The blot is one out of? similar blots.

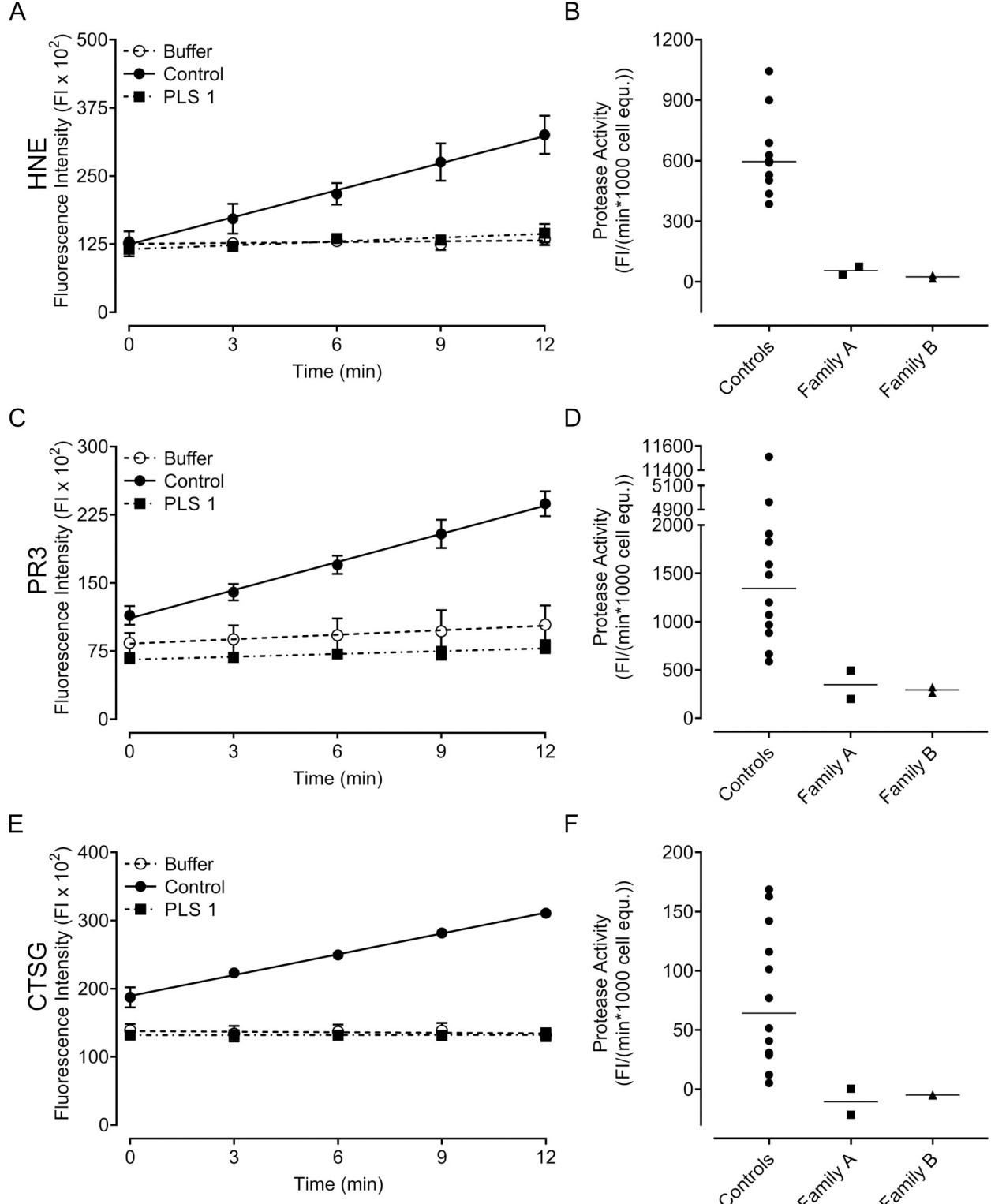

**Fig 4. Fluorescence-based protease activity assay for NSP activity of neutrophil lysates.** (A, C, E) Representative results of HNE, PR3 and CTSG activity assays, respectively, with neutrophils lysates (in triplicate) of one healthy control, PLS 1, and lysate-free buffer control. Symbols are means ± standard deviation and lines represent the linear regressions of the fluorescence measurement on the linear section of the kinetic graphs (0–12 min). (B, D, F) Protease activities of neutrophils from patients with PLS (family A, n = 2 and B, n = 2) and healthy controls (n = 12) resulting from the linear regression of the fluorescence measurement (0–12 min). Each symbol represents mean per individual (of three technical repeats) and the horizontal line represents the mean of all samples from the indicated groups.

responded normally to stimulation by translocating complement receptors from the granule membranes to the cell surface and shedding surface L-selectin (Fig 5A). The PLS neutrophils from family A responded normally with regards to *in vitro* chemotaxis towards fMLF (Fig 5B). Neutrophils from PLS 2 displayed more pronounced chemotaxis towards fMLF than PLS 1, but both responses were clearly within the range displayed by cells from healthy blood donors (Fig 5B).

The ability of the PLS neutrophils to regulate cell death was tested with the help of anti-apoptotic (LPS) or pro-apoptotic (anti-CD95) factors [42, 47]. Neutrophils from all four patients and ten healthy controls were incubated and analyzed on several occasions. In the case of PLS

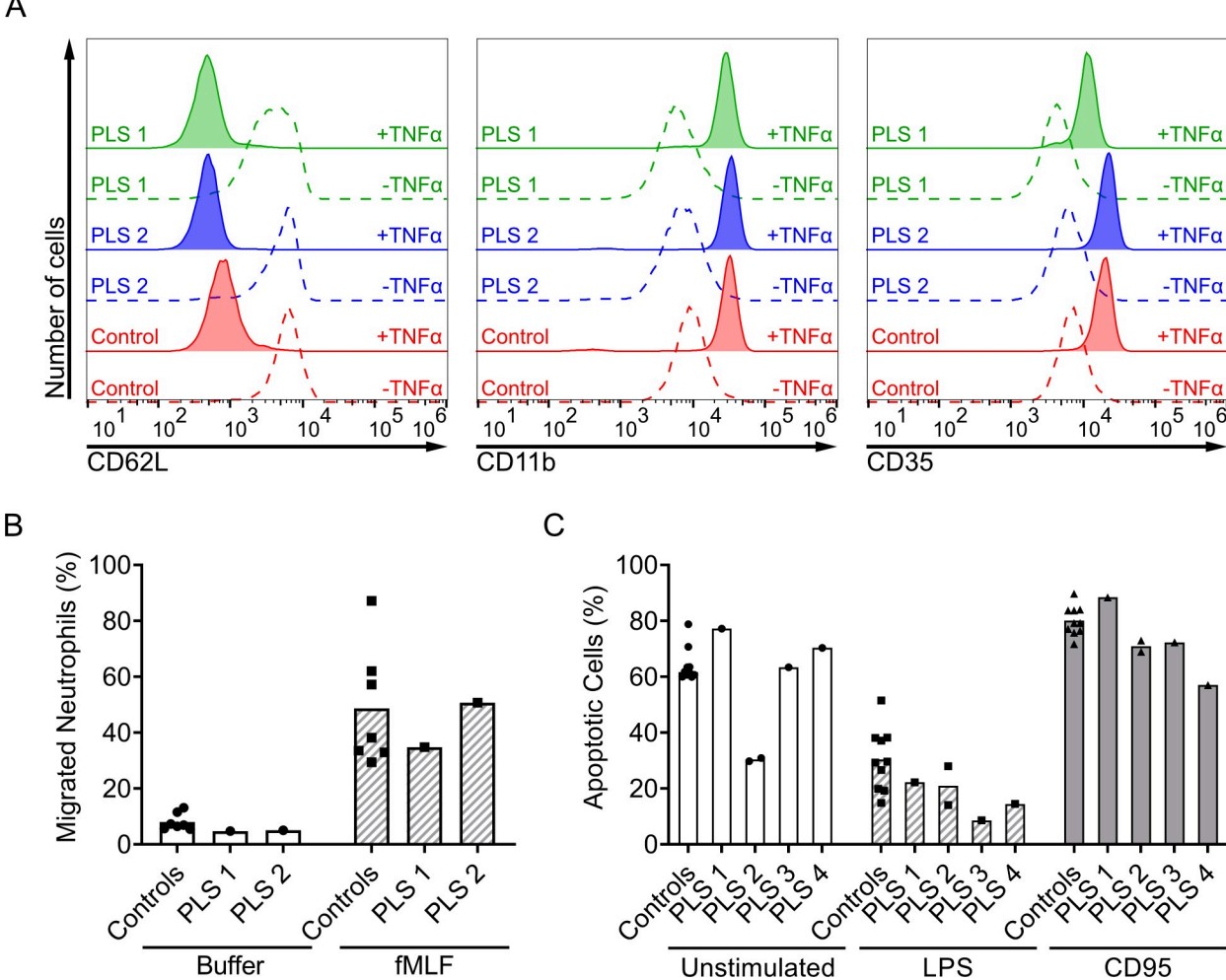

**Fig 5. Activation of neutrophils in whole blood samples, chemotaxis of neutrophils and apoptosis regulation of neutrophils.** (A) Histograms of flow cytometry analysis showing the ability of the neutrophils to regulate the surface marker expression during activation. Results shown are from the patients of family A (PLS 1 & 2) and one representative healthy control. Neutrophils were left unstimulated or activated with TNFα (10 ng/ml) for 20 min at 37˚C and labelled for extracellular markers CD62L, CD11b and CD35. (B) Bar diagram summarizing the migration for family A and healthy controls (n = 7). Isolated neutrophils were allowed to migrate through a polycarbonate chemotaxis membrane with a pore size of 3 μm towards buffer or fMLF (10 nM) for 90 min at 37˚C, 5% $CO_2$. The migrated cells were lysed, and the number of cells calculated by measuring the total amount of MPO-activity per well relative to a well corresponding to the total amount of cells. (C) Diagram showing the percentage of apoptotic cells after incubation with indicated stimulus for patients of family A (experiment for PLS 2 performed twice on different occasions), family B, and healthy controls (n = 10). Isolated neutrophils were placed in cell-culture buffer substituted with anti-apoptotic LPS, pro-apoptotic CD95 or left unstimulated for 20 h of incubation at 37˚C, 5% $CO_2$ before measurement by flow cytometry. Cells that stained positive for Annexin-V but negative for 7-AAD were classified as apoptotic.

2, we had the opportunity to perform the experiment on two separate occasions. From this limited data we conclude that the PLS neutrophils respond normally with decreased apoptosis in response to LPS and increased apoptosis when stimulated with anti-CD95 (Fig 5C). Only the cells from PLS 2 seem to display a lower level of spontaneous apoptosis, below the range observed in the 12 healthy controls, and these cells also did not respond to the same extent to anti-apoptotic LPS as did neutrophils form other individuals (Fig 5C). Importantly, the suppressed spontaneous neutrophil apoptosis displayed by PLS 2 was not observed for any other patient with PLS of neither family A nor B.

PLS neutrophils were able to mount a seemingly normal respiratory burst upon stimulation with PMA; both when measuring extracellular release of ROS and when measuring ROS produced intracellularly. Based on our earlier findings [48], we know that the ROS produced by neutrophils varies greatly in the general population; with this in mind we conclude that the radical responses of the PLS neutrophils fall within the range of the biological variation (Fig 6).

Finally, we investigated *in vitro* NETs release in response to PMA stimulation, a functional response reported to be altered in PLS neutrophils as it is dependent on HNE activity [11]. Control neutrophils typically form NETs from 2 hours after stimulation, but up until 3 hours of incubation with PMA only very few PLS neutrophils from family A had produced NETs as visualized microscopically (Fig 7A). The Sytox green assay (Fig 7B) confirmed this observation, showing that PMA-triggered NET formation was virtually absent <3 h. This is in line with previous reports on defect PMA-triggered NET formation by PLS neutrophils [9, 10]. However, upon prolonged incubation, PMA stimulated PLS neutrophils from PLS 1 and 2 in fact released significantly (*p* = 0.044, Paired Student's t-test, PLS unstimulated vs. PLS+PMA, >3h, n = 2) more NETs than unstimulated PLS cells (Fig 7B). The experiment was performed on two separate occasions, for two different healthy controls, once for PLS 1 and twice for PLS 2. These data show that even though PMA triggered NETs release is severely hampered, the process is not completely absent in PLS neutrophils.

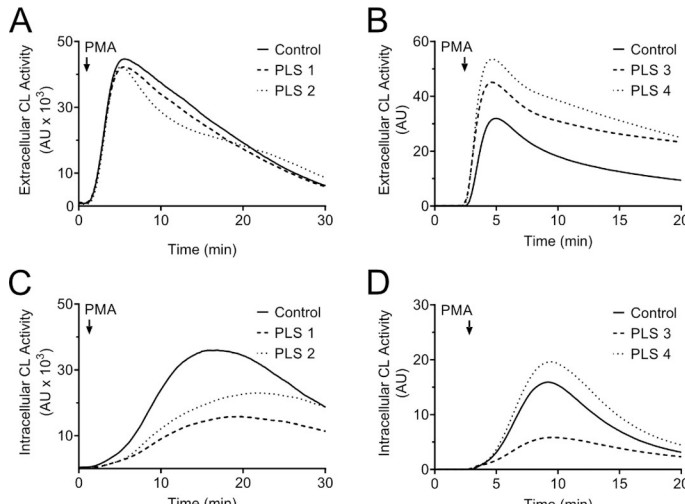

**Fig 6. Extra- and intracellular ROS responses upon PMA stimulation.** (A, B) Isolated neutrophils of patients from family A and B as well as one healthy control were suspended in buffer substituted with Isoluminol and HRP and stimulated with PMA. The extracellular ROS production was recorded as chemiluminescence (CL) emitted by the system. (C, D) Isolated neutrophils of patients from family A and B as well as one healthy control were suspended in buffer substituted with luminol and SOD+catalase and stimulated with PMA. The intracellular ROS production was recorded as CL emitted by the system.

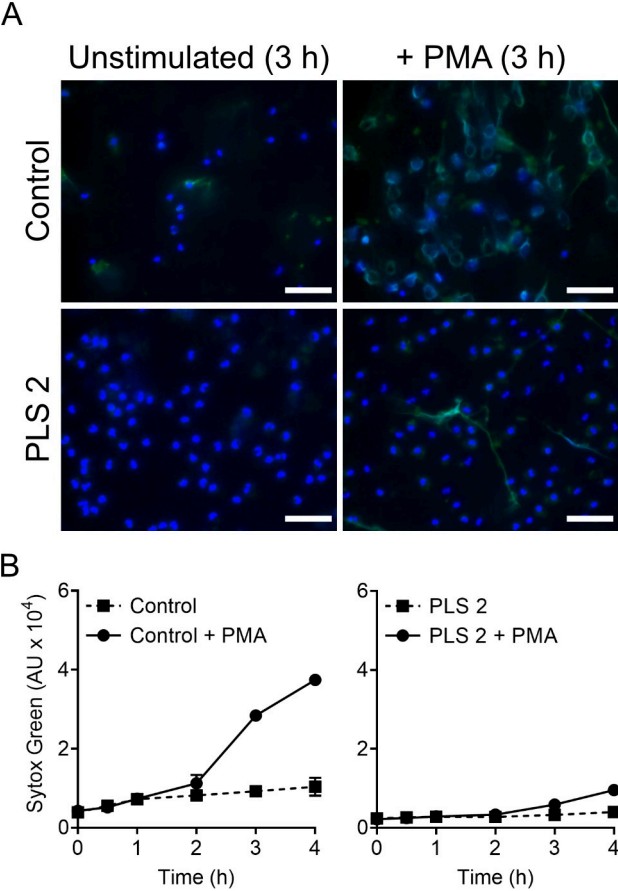

**Fig 7. NETosis of neutrophils upon PMA stimulation.** (A) Micrographs showing neutrophils on coverslips after 3 h of incubation. Isolated neutrophils were allowed to settle on coverslips and incubated with a cell culture buffer with or without PMA (50 nM) for 3 h at 37˚C, 5% $CO_2$. After fixation, the cells were stained with the DNA-dye DAPI (blue) and anti-MPO stain (green) to highlight the formation of NET-structures. Magnification, 40x. Scale bars, 50 μm. Representative images are shown for PLS 2 and one healthy control with or without PMA stimulation. (B) Graphs showing the amount of extracellular DNA during incubation of neutrophils with PMA. Representative results for PLS neutrophils of family A (PLS 2) and one healthy control. Isolated neutrophils were suspended in cell culture medium with the DNA-dye Sytox Green (1.25 μM), with or without PMA (50 nM). The cells were cultured for 4 h at 37˚C, 5% $CO_2$ and fluorescence intensity from extracellular DNA was recorded regularly. Means ± standard deviations of triplicates are shown.

## Discussion

In this study we characterized a rare *CTSC* mutation in two siblings with PLS from different aspects, e.g., protease activity, neutrophil structure and function. With the help of whole genome sequencing on the patients of family A we identified a missense mutation in the *CTSC* gene, more specifically a 503A>G transversion in exon 4 predicted to cause an amino acid substitution at position 168 from tyrosine to cysteine. In databases linking mutations to human disease, this mutation was not previously reported. We also did not find any reported homozygous individuals in genome databases such as gnomAD. Beyond the siblings described here, we are only aware of one patient with PLS homozygous for this exact mutation [9]. Besides a control cohort of healthy volunteers, we had the opportunity to compare the patients of family A to patients of another family also suffering from PLS with a 755A>T mutation in the *CTSC gene*, a mutation earlier described in Toomes *et al.* and Hewitt *et al.* [7, 46]. Over 50 different types of mutations of the *CTSC* gene have been classified as PLS [10], identified as,

e.g., missense, nonsense or acceptor splice site mutations [7]. These mutations have been reported to either lead to absence or inactivation of the CTSC protease [10], ultimately leading to activity levels unable to maintain maturation of the NSPs. It is generally accepted that PLS is not an immune deficiency in the classical sense as an increased susceptibility to opportunistic infections is not widely observed [49, 50]. This was also the case in our patient group, except for one patient in family B that had one reported episode of a purulent skin cyst.

As expected in patients with PLS, neither CTSC activity nor downstream NSP activity was detected in neutrophil lysates and there were no differences between cells from family A and family B. Furthermore, no CTSC protein was found in neutrophil lysates from family A. This indicates that the 503A>G mutation indeed is a complete loss-of-function mutation that does not result in a stable, functional protein. Neutrophils with a 503A>G mutation, identical to that found in family A, have been studied previously [9] and found to lack the CTSC and the NSPs. This fits well with the complete absence of CTSC protein and NSP activity reported here. When quantifying protease activities of the healthy volunteers in our control cohort, a wide interindividual variation was observed for CTSC as well as for NSP activity. Whether this substantial variation in, e.g., PR3 activity among healthy control neutrophils (Fig 4C and 4D) is the result of differences in cellular PR3 amounts, different PR3 variants, or levels/activity of endogenous PR3 inhibitors we cannot say, but it is intriguing that a set number of neutrophils from healthy blood donors can display such wide variation in terms of protease activity. Regardless of this wide variation among healthy controls, it was clear that PLS samples stood out and lacked measurable protease activities altogether. In fact, PLS lysates were most often indistinguishable from cell-free buffer controls and sometimes displayed readings even lower than buffer controls.

Like earlier described PLS mutations, the neutrophils from family A were normal in abundance and appearance. One recent study that investigated neutrophil function in patients with PLS found that patient cells displayed reduced velocity and chemotactic accuracy towards fMLF, as compared to control neutrophils [10]. In another paper the same authors found that neutrophils from chronic non-PLS periodontitis also present with lower speed, velocity and directional accuracy towards fMLF as compared to healthy controls [51], which makes it hard to determine if it is the lack of NSP activity or the periodontal disease activity that affect the directional migration of PLS neutrophils. In our hands, the net-migration of PLS neutrophils was well within the range displayed by healthy control neutrophils. Protease activities as well as the magnitude of functional responses of primary neutrophils of healthy donors showed substantial interindividual variation. This remarkable variation has been previously described, e.g., regarding the magnitude of ROS production [48]. That points to the importance of including a substantial cohort of healthy controls when testing whether patient cells function abnormally and is especially important when studying rare diseases where the size of the patient groups is naturally limited. PMA-triggered ROS production of PLS neutrophils have been previously reported to be increased as compared to control neutrophils; a conclusion based on data from four patients and three healthy controls [10]. We did not observe any clear-cut differences regarding ROS production and when comparing our patient data with a larger set of controls [48] it was clear that all four PLS samples were well within the normal range of responses. Based on our functional assays, we observed no clear-cut differences that were common for all patients with PLS, or for both patients of one family. Thus, these data indicate that lack of CTSC and NSP activity does not alter neutrophil priming, chemotaxis towards fMLF, regulation of apoptosis or PMA-induced ROS production. Among these assays, only the spontaneous neutrophil apoptosis displayed by PLS 2 was outside the range of healthy controls. This feature was however not shared by any other patient with PLS and is thus likely not the direct result of abolished protease activity. It is possible that the prolonged and severe

peri-implantitis experienced by PLS 2 is reflected as systemically suppressed spontaneous neutrophil apoptosis as is the case during, e.g., sepsis and autoinflammation [47, 52]. In previous work, it has been speculated that an overall heightened anti-apoptotic milieu already leads to a prolonged life span in neutrophils, thus anti-apoptotic stimulants like LPS cannot reduce the occurrence of spontaneous apoptosis further [47, 53]. This would fit well with the weak response to LPS displayed by neutrophils from PLS 2.

One neutrophil function described by many to be dependent on HNE activity, and thus defective in PLS neutrophils, is NET formation in response to PMA stimulation [10, 11]. In line with this, we observed a potently delayed as well as reduced NET formation by neutrophils from all patients with PLS. Our previous work showed that PLS neutrophils (from PLS 2) are perfectly able to form NETs when stimulated with the membrane-disturbing protein PSMα2 [40] and it is thus clear that NETs may be induced in both HNE-dependent and independent ways. Thus, it is more correct to say that PLS neutrophils are deficient in terms of PMA-triggered NET formation, as opposed to being deficient for NET formation in general. The importance of NETs, both ROS or HNE dependent, for human health is not clear, but it is interesting to note that though both CGD and PLS are deficient for PMA-induced *in vitro* NETs, only CGD is characterized by frequent infections. This observation could be taken in support of the view that the susceptibility to infections in CGD has little to do with the inability of CGD neutrophils to form NETs in response to PMA. It is also unlikely that a dysfunctional NETs response in PLS is linked to the periodontal pathology of these patients, since CGD is not typically characterized by periodontal inflammation. Furthermore, PMA-triggered NET formation is also dependent on intracellular ROS production and MPO activity [54, 55] and since these processes were not dysfunctional in PLS neutrophils, the lack of HNE activity is the likely explanation to the reduced levels of PMA-induced NETs formed by PLS neutrophils. Importantly, even though PLS neutrophils displayed reduced NET formation in response to PMA, significant NET formation was in fact detected upon longer incubation times (>3 h) in comparison to PMA-free controls. Thus, HNE activity is a strong influence on the production of PMA-triggered NETs, but it is not indispensable. However, we cannot completely rule out that residual HNE activity in the PLS neutrophils, below the detection limit of our assay, can explain the fact that PLS neutrophils form some NETs after being stimulated with PMA.

In conclusion, in this study we demonstrate that the 503A>G *CTSC* mutation identified in family A compares to previously described PLS mutations in terms of aggressive periodontal pathology and characteristics of neutrophil functions. The changes in neutrophil function caused by *CTSC* mutations seem limited to the absence of protease activity and the diminished formation of NETs upon PMA stimulation.

## Supporting information

**S1 Raw images.**
(PDF)

## Acknowledgments

We thank Stina Lassesson from the Core Facility at the Sahlgrenska Academy for performing the whole genome sequencing. Furthermore, we would like to thank Katarina Truvé from the Bioinformatics Core Facility at the Sahlgrenska Academy for analysis of the sequencing data.

## Author Contributions

**Conceptualization:** Felix P. Sanchez Klose, Johan Bylund.

**Data curation:** Felix P. Sanchez Klose, Halla Björnsdottir, Agnes Dahlstrand Rudin, Tishana Persson, Arsham Khamzeh, Martina Sundqvist, Sara Thorbert-Mros, Régis Dieckmann, Karin Christenson.

**Funding acquisition:** Johan Bylund.

**Investigation:** Felix P. Sanchez Klose, Halla Björnsdottir, Agnes Dahlstrand Rudin, Tishana Persson, Arsham Khamzeh, Martina Sundqvist, Sara Thorbert-Mros, Régis Dieckmann, Karin Christenson.

**Methodology:** Felix P. Sanchez Klose, Karin Christenson, Johan Bylund.

**Supervision:** Karin Christenson, Johan Bylund.

**Validation:** Felix P. Sanchez Klose.

**Writing – original draft:** Felix P. Sanchez Klose, Karin Christenson, Johan Bylund.

**Writing – review & editing:** Felix P. Sanchez Klose, Halla Björnsdottir, Agnes Dahlstrand Rudin, Martina Sundqvist, Sara Thorbert-Mros, Régis Dieckmann, Johan Bylund.

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
