## [Decision Letter · Decision Letter 0]

4 Aug 2021

PONE-D-21-22412

A rare CTSC mutation in Papillon-Lefèvre Syndrome results in abolished serine protease activity and reduced NET formation but otherwise normal neutrophil function

PLOS ONE

Dear Dr. Sanchez Klose,

Thank you for submitting your manuscript to PLOS ONE. After careful consideration, we feel that it has merit but does not fully meet PLOS ONE’s publication criteria as it currently stands. Therefore, we invite you to submit a revised version of the manuscript that addresses the points raised during the review process.

Specific revisions must address issues raised by reviewers, especially:

- CTSC protein levels in neutrophils by western blot or flow cytometry including potential differences between families A and B.

- changes in figures (dot plots instead of box/whiskers) and in manuscript text as suggested by reviewers

-better delineate the effect of new mutation on NET by comparing NET release induced by other physiological agonists and in comparison with other patients with other PLS mutations.

We look forward to receiving your revised manuscript.

Kind regards,

Charaf Benarafa, D.V.M., Ph.D.

Academic Editor

PLOS ONE

Journal Requirements:

Reviewers' comments:

Reviewer's Responses to Questions

**Comments to the Author**

1. Is the manuscript technically sound, and do the data support the conclusions?

Reviewer #1: No

Reviewer #2: Yes

Reviewer #3: Yes

2. Has the statistical analysis been performed appropriately and rigorously? 

Reviewer #1: No

Reviewer #2: N/A

Reviewer #3: Yes

3. Have the authors made all data underlying the findings in their manuscript fully available?

Reviewer #1: No

Reviewer #2: Yes

Reviewer #3: Yes

4. Is the manuscript presented in an intelligible fashion and written in standard English?

Reviewer #1: Yes

Reviewer #2: Yes

Reviewer #3: Yes

5. Review Comments to the Author

Reviewer #1: In this manuscript the authors examined two siblings with PLS due to a rare mutation in CTSC, 503A>G and compared them to another family of PLS previously described. Although this mutation is unusual, it has been previously described (ref 9). Also, one of the siblings, PLS2’s ability to form NET was previously reported (ref 41). Otherwise, the authors suggested that the neutrophil morphology, priming, chemotaxis, ROS production, and regulation of apoptosis were unchanged but NET formation upon PMA stimulation was severely depressed. These phenotypes were previously reported in PLS patients. PLS2, however, displayed more pronounced chemotaxis towards fMLF than PLS1 and lower level of spontaneous apoptosis. Given that most neutrophil phenotypes are similar to previously described PLS mutations, it would have been interesting to understand why PLS2 behaves differently. Otherwise, there is no or minimal gain in understanding of PLS offered by this manuscript.

Other points:

- Why not present NSP activities for all four PLS patients individually instead of box and whisker plots for each family?

- Also, why use box and whisker plots for figure 3 but not for figure 2?

- In figure 4, why was comparison to PLS3 and 4 not presented for all experiments?

- In figure 6, why was NET formation shown only for PLS2? Do PLS1, 3, and 4 neutrophils have the same delayed response to PMA as PLS2? What about NET formation to PSMa2?

Reviewer #2: Klose and coworkers described and analyzed four PLS patients from two families (A,B) with two different homozygous mutations in the cathepsin C gene. The mutation found in family A is an extremely rare allele deposited in some databases, and only one individual with homozygosity for the same Y168C residue substitution has been discovered in a previous paper. The second variation, Q252L, was reported to occur in one affected member of another family 20 years ago, but two other missense changes were also found in the CTSC gene of this individual. Hence the important work of Klose et al. underscores the clinical significance of these homozygous changes in the CTSC sequence and presents a variety of functional and biochemical data for neutrophils derived from new patients.

To improve the manuscript, the following points should be addressed.

Line 21: replace “indicated to be involved” by “suggesting its involvement in a …”

The statement refers to CTSC and its impact on neutrophil functions. The three NSPs may be present in active form at low levels in some neutrophil populations and could be converted by other enzymes during maturation or after release.

Line 22: delete “however” and “consequentially”

Line 44: Homozygosity of CTSC mutations is frequently observed in consanguineous families and in small ethnic populations with a high degree of inbreeding. Other genetic variations may modify and contribute to the clinical manifestations and the functional impairments of neutrophils. To explain all manifestations as the result of a monogenic disease is an oversimplification.

Line 59: insert “to” killing and to degrade

Line 63: It is inappropriate to equal cathepsin G (Ctsg) and elastase (Ela2) deficient mice with PLS patients. Double deficient mice do not acquire infections spontaneously. The differences observed between wild type and double deficient mice are only seen under extreme experimental conditions titrating these mice with increasing doses of bacteria until they die. The authors should compare PLS patients with Ctsc-deficient (Ctsc-null) mice which are healthier than affected humans and do not suffer from periodontitis. Azurocidin is completely absent in all rodents by contrast to humans and may play a human-specific function which missing in PLS. The state of the research should be reported correctly.

Line 448: Ela2-deficient or CTSG-deficient mice are not more susceptible to inflammations and infections than wild type mice under normal housing conditions. Differences are only seen under extreme experimental challenges. This interpretation of animal models gives a wrong impression to your readers, I am afraid.

Line 506: please modify this sentence to make its content clear “how important NET formation induced by persistent high ROS levels and executed by HNE is in vivo

Line 514: the authors are obviously very familiar with the JCI paper of Sorensen OE et al. from 2014, in which eosinophil peroxidase was found to be strongly reduced in PLS neutrophils (supplemental table). Have they indeed convinced themselves independently that peroxidase activities and peroxidase levels were unaltered in the neutrophils of the patients they have studies?

Reviewer #3: In their manuscript, Sanchez Klose and collegues study a CTSC mutation that is associated with Papillon-Lefèvre Syndrome. The reported a new mutation in CTSC and characterise it’s impact on neutrophil functions. The manuscript is interesting and well-``written, and I believe interesting for Plos One readership. I have a few comments that I hope will be helpful to the authors.

Major points:

It is unclear what are the consequence of the mutation. Is this mutation affecting the stability/half-life or expression of CTSC? A simple Western Blot probing for CTSC would be very appropriate here to confirm that the protein expression is not affected by the mutation.

Figure 4C): It seems that CD95 is not increasing much apoptosis (over basal level of death in untreated cells). Would an earlier time point be more appropriate to measure cell death?

Minor points:

Line 152: What is the calcium concentration?

Figure 6A: A quantification of the microscopy slides would be relevant here. Would it be possible to report the proportion of cells generating NETs upon PMA treatment?

It would be intereting to discuss more the effect of Y168C on the protease activity. Why is it affecting the activity? Could you suggest an effect based on CTSC structure?

6. PLOS authors have the option to publish the peer review history of their article (what does this mean?). If published, this will include your full peer review and any attached files.

Reviewer #1: No

Reviewer #2: No

Reviewer #3: No

---

## [Author Response · Author response to Decision Letter 0]

28 Oct 2021

Reviewer #1: 

In this manuscript the authors examined two siblings with PLS due to a rare mutation in CTSC, 503A>G and compared them to another family of PLS previously described. Although this mutation is unusual, it has been previously described (ref 9). Also, one of the siblings, PLS2’s ability to form NET was previously reported (ref 41). Otherwise, the authors suggested that the neutrophil morphology, priming, chemotaxis, ROS production, and regulation of apoptosis were unchanged but NET formation upon PMA stimulation was severely depressed. These phenotypes were previously reported in PLS patients. PLS2, however, displayed more pronounced chemotaxis towards fMLF than PLS1 and lower level of spontaneous apoptosis. Given that most neutrophil phenotypes are similar to previously described PLS mutations, it would have been interesting to understand why PLS2 behaves differently. Otherwise, there is no or minimal gain in understanding of PLS offered by this manuscript.

Other points:

- Why not present NSP activities for all four PLS patients individually instead of box and whisker plots for each family?

Response: We have changed the graphs to individual scatter dot plots. 

- Also, why use box and whisker plots for figure 3 but not for figure 2?

Response: We have changed the plots. 

- In figure 4, why was comparison to PLS3 and 4 not presented for all experiments?

Response: The differences in presented data comes from limited availability of patients for sample donation. Also, the limited amounts of blood/PMNs retrieved restricts the experiments that can be performed on one given occasion. Furthermore, patients of family B are children, thus the sample volumes and patient access have been limited. 

- In figure 6, why was NET formation shown only for PLS2? Do PLS1, 3, and 4 neutrophils have the same delayed response to PMA as PLS2? What about NET formation to PSMa2?

Response: Thank you for that feedback, we would have liked to include more experiments as well, but the aforementioned limitations also apply here. On top of that we show only PLS 2 as the result was representative of family A. We now clarified that in the text and made it clear that also neutrophils from PLS 1 demonstrate the delayed response. Regarding NET formation through PSMa2, we have previously shown that neutrophils from PLS2 indeed form NETS in response to PSMa2, to the same extent as do neutrophils form healthy controls. In the text we refer to this work (Björnsdottir H et al., Front. Immunol., 8, 257 (2017)).

Reviewer #2: 

Klose and coworkers described and analyzed four PLS patients from two families (A,B) with two different homozygous mutations in the cathepsin C gene. The mutation found in family A is an extremely rare allele deposited in some databases, and only one individual with homozygosity for the same Y168C residue substitution has been discovered in a previous paper. The second variation, Q252L, was reported to occur in one affected member of another family 20 years ago, but two other missense changes were also found in the CTSC gene of this individual. Hence the important work of Klose et al. underscores the clinical significance of these homozygous changes in the CTSC sequence and presents a variety of functional and biochemical data for neutrophils derived from new patients.

To improve the manuscript, the following points should be addressed.

Line 21: replace “indicated to be involved” by “suggesting its involvement in a …”

The statement refers to CTSC and its impact on neutrophil functions. The three NSPs may be present in active form at low levels in some neutrophil populations and could be converted by other enzymes during maturation or after release.

Response: We have changed the sentence accordingly. 

Line 22: delete “however” and “consequentially”

Response: This has been amended.

Line 44: Homozygosity of CTSC mutations is frequently observed in consanguineous families and in small ethnic populations with a high degree of inbreeding. Other genetic variations may modify and contribute to the clinical manifestations and the functional impairments of neutrophils. To explain all manifestations as the result of a monogenic disease is an oversimplification.

Response: We strongly agree with the reviewer that patients with PLS could be a very heterogeneous group with respect to different hereditary predispositions and clinical symptoms. Therefore, it is added value that we often compare 2 distinct PLS families. We have clarified the language throughout the manuscript to make that clearer and to not overinterpret the direct influence of CTSC mutations on especially the clinical manifestations.

Line 59: insert “to” killing and to degrade

Response: Thank you, the change was made. 

Line 63: It is inappropriate to equal cathepsin G (Ctsg) and elastase (Ela2) deficient mice with PLS patients. Double deficient mice do not acquire infections spontaneously. The differences observed between wild type and double deficient mice are only seen under extreme experimental conditions titrating these mice with increasing doses of bacteria until they die. The authors should compare PLS patients with Ctsc-deficient (Ctsc-null) mice which are healthier than affected humans and do not suffer from periodontitis. Azurocidin is completely absent in all rodents by contrast to humans and may play a human-specific function which missing in PLS. The state of the research should be reported correctly.

Response: We agree that the mouse models are not reflective of the specific phenotype present in patients with PLS, and the insights from animal experiments are not easy to translate to a human setting. Therefore, we decided to remove all mentioning of murine models from our manuscript. 

Line 448: Ela2-deficient or CTSG-deficient mice are not more susceptible to inflammations and infections than wild type mice under normal housing conditions. Differences are only seen under extreme experimental challenges. This interpretation of animal models gives a wrong impression to your readers, I am afraid.

Response: We agree and we have therefore removed all mentioning of murine models from the revised manuscript. 

Line 506: please modify this sentence to make its content clear “how important NET formation induced by persistent high ROS levels and executed by HNE is in vivo

Response: The paragraph about NET formation has been changed for clarity.

Line 514: the authors are obviously very familiar with the JCI paper of Sorensen OE et al. from 2014, in which eosinophil peroxidase was found to be strongly reduced in PLS neutrophils (supplemental table). Have they indeed convinced themselves independently that peroxidase activities and peroxidase levels were unaltered in the neutrophils of the patients they have studies?

Response: Yes, we have in fact read that particular paper a number of times, but this intriguing detail had escaped our notice and we did not perform any direct experiments on peroxidase (EPO or MPO) activities and levels. Since the ROS method employed for determining intracellular ROS production is completely dependent on MPO activity (Björnsdottir H et al., Free Radical Biology and Medicine, 89 (2015)), the fact that PLS neutrophils are within the normal range of responses, indicate that at least MPO activity of the cells are normal. As we argue in the manuscript, a much bigger patient and control group would be necessary to make clear assessments on subtle differences of ROS production and peroxidase activities. 

Reviewer #3: 

In their manuscript, Sanchez Klose and collegues study a CTSC mutation that is associated with Papillon-Lefèvre Syndrome. The reported a new mutation in CTSC and characterise it’s impact on neutrophil functions. The manuscript is interesting and well-``written, and I believe interesting for Plos One readership. I have a few comments that I hope will be helpful to the authors.

Major points:

It is unclear what are the consequence of the mutation. Is this mutation affecting the stability/half-life or expression of CTSC? A simple Western Blot probing for CTSC would be very appropriate here to confirm that the protein expression is not affected by the mutation.

Response: We agree with the reviewer that it would be very interesting to figure out the concrete effect of the mutation. For that reason, we have performed the WB as suggested, and the data clearly demonstrate that protein expression is in fact severely affected by the mutation and both patients of family A lack CTSC protein. We have now added one representative blot to the manuscript (new figure 3) and discuss these data in the result as well as discussion section of the revised manuscript.

The figure shows a western blot of recombinant human CTSC, neutrophil lysates of patients with PLS (PLS 1 and 2) as well as two healthy controls. The band for CTSC is expected around 23 kDa.

Figure 4C): It seems that CD95 is not increasing much apoptosis (over basal level of death in untreated cells). Would an earlier time point be more appropriate to measure cell death?

Response: A valid point. Shorter incubations would have been better to see CD95-induced acceleration of apoptosis, but 20 h is typically the time point where roughly half of the incubated neutrophils (without any additions) are apoptotic which enable us to detect both decrease (+LPS) and increase (+CD95) in the same experimental setup.

Minor points:

Line 152: What is the calcium concentration?

Response: The calcium concentration (1mM) has been added.

Figure 6A: A quantification of the microscopy slides would be relevant here. Would it be possible to report the proportion of cells generating NETs upon PMA treatment?

Response: For quantification, we decided to rely on the fluorescent DNA probe Sytox-green to measure the amount of NETs released over time by measuring the amount of extracellular DNA in the samples with the help of a plate reader. The microscopy slides were used to illustrate the appearance of NETs as well as the presence of MPO. By this we get both qualitative (microscopy) and quantitative (Sytox) data for NET formation. Such combination of two methods is commonly employed also by others. 

It would be intereting to discuss more the effect of Y168C on the protease activity. Why is it affecting the activity? Could you suggest an effect based on CTSC structure?

Response: An interesting point. We have tried to find structural information related to this position (168), but even though CTSC has been crystallized and structurally defined (see, e.g., Uniprot, #P53634), the amino acids around 168 are located in between two beta strands and appear to not be structurally defined. However, as based on our immunoblotting for CTSC it seems that the (mutant) protein is not expressed in neutrophils from patients with PLS (see response above and new fig 3). We can at present not state the reason why the mutated protein is absent, but this this finding of course explains why no CTSC activity was measured from these cells.

---

## [Decision Letter · Decision Letter 1]

9 Dec 2021

A rare CTSC mutation in Papillon-Lefèvre Syndrome results in abolished serine protease activity and reduced NET formation but otherwise normal neutrophil function

PONE-D-21-22412R1

Dear Dr. Sanchez Klose,

We’re pleased to inform you that your manuscript has been judged scientifically suitable for publication and will be formally accepted for publication once it meets all outstanding technical requirements.

Kind regards,

Charaf Benarafa, D.V.M., Ph.D.

Academic Editor

PLOS ONE

Additional Editor Comments (optional):

Reviewers' comments:

Reviewer's Responses to Questions

**Comments to the Author**

1. If the authors have adequately addressed your comments raised in a previous round of review and you feel that this manuscript is now acceptable for publication, you may indicate that here to bypass the “Comments to the Author” section, enter your conflict of interest statement in the “Confidential to Editor” section, and submit your "Accept" recommendation.

Reviewer #2: All comments have been addressed

Reviewer #3: All comments have been addressed

2. Is the manuscript technically sound, and do the data support the conclusions?

Reviewer #2: Yes

Reviewer #3: Yes

3. Has the statistical analysis been performed appropriately and rigorously? 

Reviewer #2: N/A

Reviewer #3: Yes

4. Have the authors made all data underlying the findings in their manuscript fully available?

Reviewer #2: Yes

Reviewer #3: Yes

5. Is the manuscript presented in an intelligible fashion and written in standard English?

Reviewer #2: Yes

Reviewer #3: Yes

6. Review Comments to the Author

Reviewer #2: (No Response)

Reviewer #3: My comments have been addressed to a reasonable extend. I have no other concerns regarding this manuscript.

7. PLOS authors have the option to publish the peer review history of their article (what does this mean?). If published, this will include your full peer review and any attached files.

Reviewer #2: No

Reviewer #3: No

---

## [Editor Report · Acceptance letter]

13 Dec 2021

PONE-D-21-22412R1 

A rare *CTSC* mutation in Papillon-Lefèvre Syndrome results in abolished serine protease activity and reduced NET formation but otherwise normal neutrophil function 

Dear Dr. Sanchez Klose:

I'm pleased to inform you that your manuscript has been deemed suitable for publication in PLOS ONE. Congratulations! Your manuscript is now with our production department. 

Kind regards, 

on behalf of

Prof. Dr. Charaf Benarafa 

Academic Editor

PLOS ONE